# Image-Based Dietary Assessment and Tailored Feedback Using Mobile Technology: Mediating Behavior Change in Young Adults

**DOI:** 10.3390/nu11020435

**Published:** 2019-02-19

**Authors:** Charlene L. Shoneye, Satvinder S. Dhaliwal, Christina M. Pollard, Carol J. Boushey, Edward J. Delp, Amelia J. Harray, Peter A. Howat, Melinda J. Hutchesson, Megan E. Rollo, Fengqing Zhu, Janine L. Wright, Iain S. Pratt, Jonine Jancey, Rhiannon E. Halse, Jane A. Scott, Barbara Mullan, Clare E. Collins, Deborah A. Kerr

**Affiliations:** 1School of Public Health, Curtin University, Perth, WA 6845, Australia; charlene.shoneye@curtin.edu.au (C.L.S.); S.Dhaliwal@curtin.edu.au (S.S.D.); c.pollard@curtin.edu.au (C.M.P.); amelia.harray@curtin.edu.au (A.J.H.); p.howat@curtin.edu.au (P.A.H.); J.Wright@exchange.curtin.edu.au (J.L.W.); J.Jancey@curtin.edu.au (J.J.); rhiannon.halse@curtin.edu.au (R.E.H.); jane.scott@curtin.edu.au (J.A.S.); 2East Metropolitan Health Service, Perth, WA 6845, Australia; 3Epidemiology Program, University of Hawaii Cancer Centre, Honolulu, HI 96813, USA; cjboushey@cc.hawaii.edu; 4Department of Nutrition, Purdue University, West Lafayette, IN 47907, USA; 5School of Electrical and Computer Engineering, Purdue University, West Lafayette, IN 47907, USA; ace@ecn.purdue.edu (E.J.D.); zhu0@ecn.purdue.edu (F.Z.); 6School of Health Sciences and Priority Research Centre in Physical Activity and Nutrition, Faculty of Health and Medicine, University of Newcastle, Newcastle, NSW 2308, Australia; melinda.hutchesson@newcastle.edu.au (M.J.H.); megan.rollo@newcastle.edu.au (M.E.R.); clare.collins@newcastle.edu.au (C.E.C.); 7Cancer Council Western Australia, Subiaco, WA 6008, Australia; SPratt@cancerwa.asn.au; 8Western Australian Cancer Prevention Research Unit (WACPRU), School of Psychology, Curtin University, Perth, WA 6845, Australia; 9Health Psychology & Behavioural Medicine Research Group, School of Psychology, Curtin University, Perth, WA 6845, Australia; Barbara.mullan@curtin.edu.au

**Keywords:** tailored feedback, image-based dietary assessment, technology intervention, text messaging, young adult, tailoring, vegetables, junk food, sugar-sweetened beverages, alcohol, mobile food record

## Abstract

Assessing the implementation of nutrition interventions is important to identify characteristics and dietary patterns of individuals who benefit most. The aim was to report on young adults’ experiences of receiving dietary feedback text messaging intervention. Diet was captured using an image-based 4-day mobile food record^TM^ application (mFR^TM^) and assessed to formulate two tailored feedback text messages on fruit and vegetables and energy-dense nutrient-poor (EDNP) foods and beverages. At 6-months 143 participants completed a second mFR^TM^ and a questionnaire evaluating the dietary feedback. Participants who agreed the text messages made them think about how much vegetables they ate were more likely to increase their intake by at least half a serve than those who disagreed [odds ratio (OR) = 4.28, 95% Confidence Interval (CI): 1.76 to 10.39]. Those who agreed the text messages made them think about how much EDNP foods they ate, were twice as likely to decrease their intake by over half a serve (OR = 2.39, 95%CI: 1.12 to 5.25) than those who disagreed. Undertaking detailed dietary assessment ensured the tailored feedback was constructive and relevant. Personal contemplation about vegetable and EDNP food intake appears to be a mediator of dietary change in young adults.

## 1. Introduction

Poor diet is associated with increased risk of diseases such as cardiovascular disease; type 2 diabetes; and some cancers, with obesity a mediating factor [1,2]. Australian guidelines recommend regular consumption of diets high in fruit and vegetables, limiting consumption of alcohol and avoiding energy dense nutrient poor (EDNP) foods high in added sugars, saturated fat and sodium to reduce the risk of diet-related diseases [3]. Dietary patterns vary throughout the life course and it is important to ensure good eating habits are maintained into young adulthood to protect against preventable disease. Compared with other age groups, Australian young adults consume the least amount of fruits and vegetables and the most alcohol and EDNP foods and beverages with more than one-third (35%) of total energy coming from EDNP food and beverages [4,5]. Consequently, young adults are gaining weight more rapidly than any other age group, leading to an earlier onset of overweight and obesity than in previous generations putting them at a greater risk of preventable diet-related disease [6]. Young adults, a group in transition from adolescence to adulthood, represent an important target group for improving dietary habits and preventing weight gain. Developing effective interventions requires an understanding of the motivators and barriers for healthy eating, as well as how best to engage young adults over time [7,8]. 

Digital communications, including email and text messaging, have been shown to effectively engage young adults in dietary interventions [9,10,11]. Digital behavior change interventions (DBCI) combining these approaches are evolving rapidly, introducing new challenges for evaluation including understanding their role in effective engagement with end users [12]. Evaluation of a text message intervention in young adults found website and mobile app engagement to be low during the study period, with participants preferring self-monitoring apps and individualized resources [13]. In a systematic review of internet-delivered weight loss interventions, personalized feedback targeting diet and physical activity behaviors appeared to be an important behavior change technique [14]. Feedback based on characteristics unique to the individual, provide the user with personalized or tailored feedback, which can be delivered as printed material, email or text message [15]. Tailoring information places less cognitive load on the individual and enhances engagement [16]. Using mobile technology in tailored dietary interventions may be a cost-effective way to engage young people in evidence-based, behavior change interventions, with potential for a population level reach. 

An important aspect of tailoring is to ensure feedback to individuals has personal relevance. Accurate and timely dietary assessment is paramount for tailored dietary interventions. Personally relevant and effective feedback can guide individuals to identify dietary changes to improve their health [17]. Collecting dietary data via mobile devices enables real-time data collection, reducing the risk of recall bias and participant burden [18]. Young adults are more willing to record their diet either online or with mobile devices compared with a written food record [19,20,21].

The Connecting Health and Technology (CHAT) study was a 6-month randomized controlled trial that evaluated the effectiveness of tailored dietary feedback and weekly text messaging to improve intake of fruit, vegetables and EDNP food and beverages in young adults (18 to 30 years) [9]. EDNP foods, colloquially referred to as ‘junk food’ in communications with participants [22,23], are those foods and beverages high in energy, saturated fat, added sugar, salt or alcohol, and low in nutritional value. The CHAT study protocol and outcomes have been previously published [9,24]. In brief, after baseline assessment, participants were randomly assigned to one of three groups: (1) combined dietary feedback and weekly text messages; (2) dietary feedback text messages; or (3) control group who did not receive any text messages. Tailored dietary feedback alone led to a decrease in consumption of EDNP foods in men and sugar-sweetened beverages (SSB) in women, and a 1.7 kg reduction in body weight [9]. However, additional weekly text messages did not appear to have any further benefit, reinforcing the need to further evaluate factors associated with the effectiveness of the dietary intervention. 

Assessing the implementation of DBCI is important in order to better understand and identify the characteristics and diet of individuals who benefitted most from tailored feedback. This will help guide future interventions design in young adults. The aim of the current study was to report on young adults’ experiences of receiving the dietary feedback following the 6-month dietary intervention and to determine whether those experiences were associated with positive improvements in dietary intake. 

## 2. Materials and Methods 

### 2.1. Study Design and Participant Recruitment

Data were collected from a population-based sample of 247 young adults (18 to 30 years) taking part in a 6-month randomized controlled trial (RCT) to evaluate the effectiveness of tailored dietary feedback and weekly text messaging support to improve diet. Only young adults randomized at baseline to dietary feedback text messaging intervention groups were included in this analysis (n = 164). The study protocol and trial outcomes have been published previously [9,24]. Participants were selected from 57 suburbs within the Perth metropolitan area in Western Australia to provide representation across socio-economic status through the Commonwealth electoral roll, a compulsory enrolment system for Australian adults. After receiving a letter of invitation, those who wished to take part underwent eligibility screening either by telephone or the study website. Exclusion criteria applied if people were unable to complete the 6-month study, undertaking extreme forms of physical activity, on a special diet, currently studying or had studied nutrition, pregnant or breastfeeding, unable to attend the study center to complete the face-to-face assessments or affected by serious illness. The study was approved by the Curtin University Human Research Ethics Committee and the Department of Health, Western Australia Human Research Ethics Committee (HR 181/2011) and all participants signed informed consent. The trial was registered with the Australian New Zealand Clinical Trials Registry (ACTRN12612000250831). 

### 2.2. Intervention: Dietary Feedback 

Dietary intake was assessed in participants using an image-based dietary assessment system known as Technology Assisted Dietary Assessment or TADA mobile food record^TM^ application (mFR^TM^) [25,26,27,28]. Participants were instructed to record their food and beverage intake using the mFR^TM^ for four consecutive days (Wednesday to Saturday) with the investigator-supplied iPod Touch (iOS6) loaded with the mFR^TM^ application. When taking an image, participants were instructed to include a provided reference object known as a fiducial marker (a checkerboard pattern of known shape, size and color) to assist with food identification and portion size estimation [27,29,30]. They were instructed to record food and beverage items not captured in an image using the iPod notes section. A week later participants attended a second baseline visit to return the iPod Touch and complete additional written questionnaires. At this visit, the researcher interviewed each participant to verify the content of the images and probe for any forgotten food and beverages. 

The trained analyst assessed the mFR^TM^ using a quality scoring of food items by food group according to the Australian Guide to Healthy Eating (AGHE) standard serves [3]. For each participant, an average serve per day was calculated for fruits, vegetables, EDNP foods and beverages. Once scoring was complete, two tailored dietary feedback text messages were sent to the intervention participants one week apart, with one message for fruits and vegetables and the other for EDNP food and beverages. A standard message template previously described [9]; was used for each dietary feedback text message but personalized for each participant according to the results of the dietary analysis. Briefly, for the fruit and vegetable messages, participants received a message based on three levels of intake. Low intake was considered 0 to < 3.5 servings of fruits and vegetables; medium was 3.5 to < 7 servings of fruits and vegetables; at least 2 servings of fruits and 5 servings of vegetables per day met the recommendations. For example, for a low intake, “Hi Jane, it’s Kate from CHAT with your feedback. So how did you score? Ave fruit serves = 0. 5, ave veg serves = 0.5. Your fruit serves varied from 0–1, veg from 0–1 over 4 days. What’s the goal again? 2 fruit and 5 veg a day. You can only go up from here!”. For EDNP serves, participants received a message based on three levels of intake and personalized to the participant’s dietary intake. For EDNP serves of 3 or more per day, the message was personalized with key sources of EDNP serves identified from the mFR^TM^. For example, “Hi Pete, It’s Kate from CHAT with your junk food score. Ave serves = 5, varying from 4-8 over 4 days. Junk foods are fatty or sugary foods that are high in calories. So try to only eat these foods sometimes and in small amounts. Could you try eating less fatty foods e.g., pies and sweet biscuits?”. A low intake of EDNP serves (0–3 serves daily serves) included the text “looks like you are on the right track”. The language and tone of voice of the dietary feedback messages were constructed from message preference testing with focus groups [22], with an autonomous supportive style of communication [31]. 

### 2.3. Outcome: Changes in Dietary Intake Suggesting Benefit from Intervention

All participants undertook a 4-day mFR^TM^ at baseline and at the end of the intervention and were analyzed as described in Section 2.2. Participants were assessed to have benefited from the intervention if they increased their intake of vegetables or fruit by half a serve per day, or decreased their intake of EDNP foods, sugar-sweetened beverages (SSB), or alcohol by half a serve per day.

### 2.4. Participant Experiences with Dietary Feedback 

Post-intervention, participants completed a 13 item written questionnaire with 5-point Likert scales ranging from ‘strongly agree’ to ‘strongly disagree’ to measure participants’ agreement with statements concerning their perception of the dietary feedback text messages. Examples of questions relevant to this paper were he text message on my diet: (1) Told me things I did not know about my diet and what I eat, (2) Told me things about my diet I already knew, (3) Were useful in helping me to understand my diet, (4) Helped to motivate me to change my diet, (5) Made no difference to my motivation to change my diet, (6) Made me feel better about my diet, (7) Made me feel worse about my diet, (8) Made me think about the foods I eat but only for a short while, (9) Made me think about how much fruit I eat, (10) Made me think about how much vegetables I eat, (11) Made me think about how much junk food I eat, (12) Made me think about how much soft drink and sugary drinks I have. To further explore young adults’ experiences of receiving dietary feedback, four additional open-ended questions asked: (1) List what you liked most (if anything) about the feedback on your diet?; (2) List what you liked least (if anything) about the feedback on your diet; (3) Is there anything else about your diet you would have liked feedback on; and (4) additional comments on was the short feedback you received with the text messages sufficient? These open-ended comments and free text responses were imported verbatim into NVivo 12. Qualitative data were coded and patterns identified using thematic analyses by three researchers (D.A.K., C.L.S., C.M.P.) independently. Descriptive labels were then applied to categorize information into themes. The researchers met and reviewed together the findings to confirm key themes. Discussion and revision of themes were made where required.

### 2.5. Analyses 

Frequencies of responses to the tailored feedback questionnaire were collated for each question and responses categorized. Reference values were derived using the 5-category Likert response scales used in the dietary feedback questionnaires. These were recorded as agreed (strongly agree and agree) or, neutral and disagree (strongly disagree and disagree). Logistic regression was used to analyze the change in food group serves (by 0.5 serves) from baseline with participants’ perceptions on whether the dietary feedback text messages made them think about consumption of vegetables, fruit, EDNP, SSB or alcohol. The results were adjusted for age and sex. Preliminary analysis revealed that BMI, ethnicity, education level, and socioeconomic status were not associated (*p* > 1) with a change in food group serve, hence were not included in the multivariable model. Statistical software SPSS version 22 (SPSS Inc., Chicago, IL, USA) was used for all analyses. 

## 3. Results

### 3.1. Participants Experiences with the Dietary Feedback 

Participant characteristics at baseline are shown in Table 1. Of the 164 participants who consented to participate in the CHAT study, 143 (87%) completed two 4-day mFR^TM^ (baseline and at 6-months) and post-intervention feedback questionnaires. 

Table 2 shows participants’ perceptions regarding dietary feedback. Approximately 62% of participants agreed the text messages were useful in helping them to understand their diet and approximately half (52%) agreed the text messages helped to motivate them to change their diet. Thirty per cent of participants reported the text messages made them feel worse about their diet. More women (52%) agreed the text messages made a difference to their motivation, compared with 33% of men (*p* < 0.05). The majority of participants agreed the text messages encouraged them to think about how much fruit, vegetables and EDNP food they consumed (67%, 71%, and 65% respectively). Only 20% thought text messages made them think about how much alcohol they drank. Only 13% of participants felt the intervention provided novel information, but still found this useful (Table 2). Men were more likely than women to report that they learnt something they did not already know (21.7% compared to 8.3%, *p* < 0.05). In response to the question if the short dietary feedback was sufficient, 47% thought it was sufficient, whilst 47% wanted more feedback (remaining 6% unsure). 

### 3.2. Perception of Dietary Feedback Text Message and Dietary Intake

Table 3 reports logistic regression analysis relating participants’ perception of the text message dietary feedback to the actual change in food groups serves. Participants who agreed that the text messages made them think about how much vegetables they ate were more likely to increase their vegetable intake by more than half a serve than those who disagreed (OR = 4.28, 95% CI: 1.76–10.39, *p* = 0.001). These participants were more likely to reduce their intake of EDNP food (OR = 2.78, 95% CI: 1.28–6.04, *p* = 0.010). Participants who agreed that text messages made them think about ‘how much junk food’ they ate were more likely to decrease their EDNP food by more than half a serve (OR = 2.47, 95% CI: 1.12–5.25, *p* = 0.025). All associations were independent of age, sex and BMI.

### 3.3. Responses to Open-Ended Comments on Dietary Feedback

Participants’ responses to the four open-ended questions regarding the dietary feedback, were coded into themes. Table 4 shows examples of responses to the four open-ended questions about the dietary feedback text messages. Of the 143 participants, 103 provided comments to ‘List what you liked most about the feedback on your diet’; 75 to ‘List what you liked least about the feedback on your diet’; 91 to ‘Is there anything else about your diet you would have liked feedback on?’; and 36 provided additional comments on ‘Was the short feedback you received with the text messages sufficient?’. The dietary feedback messages were viewed positively by participants and five emerging themes were identified (Table 4). What participants appeared to like most about the text messages were that they made them think more about their diet and encouraged and motivated them to change their dietary behaviors. Participants also valued that the messages were personal and specific to them. Many said this personalized approach was important for their motivation to change. Messages were described as constructive and helpful. Some participants were shocked and surprised by the feedback. 

When asked to comment on what they liked least about the text messages, some participants found the feedback confusing and vague. They also commented they would have liked more detailed feedback. This is consistent with requests for more detailed dietary feedback when asked ‘Is there anything else about your diet you would have liked feedback on?’. Others participants, however, thought the feedback was “short and to the point”. 

## 4. Discussion

This six-month RCT evaluated young adults’ experiences of receiving the dietary feedback following a 6-month text-messaging intervention. A key finding of intervention was that contemplation about vegetable and EDNP food intake appears to be an important mediator of dietary change in young adults. Participants who agreed that dietary feedback made them think about their eating behaviors, were more likely to improve their diet during the intervention period. Those participants who agreed thinking about how much vegetables they ate were four times more likely to increase their vegetable intake by more than half a serve per day than those who disagreed. In addition, participants who agreed that text messages made them think about how much ‘junk food’ they ate were twice as likely to decrease their EDNP food by greater than half a serve. An important aspect of the intervention was the inclusion of a detailed dietary assessment using an mFR^TM^. This ensured the tailored feedback was constructive and relevant to the individual; features that appeared to be valued by the participants. 

Findings of the current study suggest young adults who believe healthy eating is important and they themselves have a healthy diet experience cognitive dissonance when presented with contrary dietary feedback to what they were expecting. Cognitive dissonance suggests individuals experience a psychological state of discomfort when holding conflicting attitudes or beliefs, which may lead to a change in behavior to reduce that discomfort [32]. This, in turn, may have driven the observed improvements in dietary intake. According to the self-determination theory (SDT) used to inform the framework underpinning the CHAT intervention, autonomous motivation is a positive predictor of long term behavioral change [33]. SDT distinguishes the different types of motivation. For instance, more autonomously motivated individuals are more likely to engage with a given behavior because it is enjoyable whereas in controlled motivation people may feel pressured to engage in the behavior for social approval or to avoid guilt [34]. Applying SDT, a cross-sectional study of nearly 3,000 US adults found autonomous motivation and perceived social support were associated with increased fruit and vegetable intake [35]. This finding emphasizes the importance of providing personally relevant dietary feedback, that can assist people to identify for themselves the dietary changes most likely to improve their health [36]. An important aspect of SDT [37] embedded in the CHAT intervention was to provide relevant dietary feedback for the person to use in making informed dietary choices.

In a systematic review of lifestyle interventions for preventing weight gain in young adults, Hebden et al. [38] recommended future trials include dietary self-monitoring and tailored feedback to increase the personal relevance to the individual. Dietary self-monitoring has been shown to be an effective behavior change strategy by raising a person’s awareness of what they are eating [39]. With mobile technology now readily accessible, together with the level of interest in mobile technology amongst young adults, collecting dietary intake data using mobile devices may lead to improved cooperation to record diet in this age group. Most dietary interventions have based tailored feedback on brief instruments that use only a few questions to assess diet rather than more detailed dietary records limiting the type and quality of feedback that can be provided to the participant [40]. A systematic review of dietary assessment methods used to evaluate interventions found that dietary components, such as fruits, vegetables, SSB and fast food, were most often assessed by single questions or brief instruments [41]. The findings of this study emphasize the importance of undertaking a detailed dietary assessment to ensure the personal relevance of the feedback.

Participants who thought about their vegetable intake as a result of receiving dietary feedback were more likely to reduce both EDNP foods and increase vegetable intake. Previous studies have reported an association between increasing consumption of vegetables and a reduction in consumption of EDNP food and SSBs [42,43]. Our results suggest this association may be mediated by intervention features that prompt individuals to think about their vegetable intake. 

Young adults in the CHAT intervention appeared to be shocked and surprised about the feedback on their dietary intake. For example, a comment from a young women “I was slightly shocked about my junk food consumption and very happy to receive the feedback”. This implies a gap between a participant’s perception of their own dietary intake and what they recorded from the 4-day mFR^TM^. This perception may be derived, in part, from a lack of knowledge and may be a barrier to change as young adults may believe their diet to be healthier than it is. 

This over-optimistic perception of young adults is evidenced by the low intake of fruits and vegetables. The median daily intake was 120 g (0.8 serves) for fruit and 135 g (1.8 serves) for vegetables; much lower than the recommended two serves of fruit and five serves of vegetables per day [3]. This is similar to Australian population data of 18–34-year-olds where a median intake of 1.3 serves of fruit and 2.1 serves of vegetables was observed [44]. In the current study, a median intake of 4.1 serves daily of EDNP food and beverages was reported. This is equivalent to more than 2400 kJ per day. A cross-sectional analysis revealed young adults who perceived their diet to be low in EDNP foods consumed less EDNP food than their peers, nevertheless their daily intake was 2.8 serves or 1,700 kJ per day [45] and inconsistent with dietary guidelines [3]. Compared with the general population for this age range, men and women in the current study consumed fewer serves of fruit and vegetables per day and had a lower BMI [4]. Such observations are not unusual, with other studies reporting similar dietary patterns among those who report a concern about personal dietary choices [46]. 

Most studies to date, have based participant dietary feedback on short questions rather than more detailed dietary records [41]. A major strength of this study is the collection of dietary intake using a 4-day mFR^TM^ which provided a more detailed and personalized measure of dietary intake. This enabled an evaluation of whether young adults’ experiences of receiving the dietary feedback were associated with positive improvements in dietary intake. Of note, there are some limitations to this study. Diet was assessed using a 4-day mFR^TM^ at baseline and 6-months and these data may not be representative of dietary intakes throughout the intervention period. Participants may have misreported their dietary intake by either not capturing all food and beverages consumed or modifying their intake during the recording period [47]. The current study did not include measures of autonomy and self-regulation. To further understand motivations towards changing dietary behaviors, future studies should study these motivational processes when planning dietary interventions. This should include examining autonomy for nonadherence in young adults whilst being respectful of their dietary choices [34]. 

## 5. Conclusions

Assessing participants’ view on various intervention components such as importance, motivational impact and frequency of communication provides useful insights for future health promotion interventions. Findings of the current study show the complexity of an individual’s perceptions, beliefs and behavior in relation to changing dietary behaviors in young adults. The effectiveness of the intervention appears to be a result of prompting, with participants encouraged to think about their intake of fruit, vegetables, EDNP food and beverages. Using text messages, together with the mFR^TM^ dietary assessment may be an effective approach for increasing motivation and awareness of dietary behavior. For young adults, text messages that provided dietary feedback were integral to dietary change. Contemplation about fruit, vegetable, EDNP food intakes appears to be an important mediator of dietary change in young adults. This study makes an important contribution to the evidence base, providing qualitative and quantitative insights into the participants’ experience of the intervention and mediators of behavior change.

## Figures and Tables

**Table 1 nutrients-11-00435-t001:** Baseline characteristics of study participants randomized to receive the dietary feedback text messages (n = 164).

Variable	Male (n = 57)	Female (n = 107)	Total (n = 164)
**Mean ± SD**			
Age (years)	24.4 ± 3.3	23.8 ± 3.3	24.0 ± 3.3
Body mass (kg)	77.4 ± 14.3	64.8 ± 15.3	69.2 ± 16.1
Height (cm)	177.7 ± 7.6	164.3 ± 6.7	169.0 ± 9.5
Body Mass Index ( BMI; kg/m^2^)	24.4 ± 4.0	24.0 ± 5.8	24.2 ± 5.3
**BMI categories (%)**			
BMI ≤ 18.5	7 (12.3%)	12 (11.2%)	19 (11.6%)
BMI > 18.5 < 25	25 (43.9%)	65 (60.7%)	90 (54.9%)
BMI ≥ 25 < 30	21 (36.8%)	17 (15.9%)	38 (23.2%)
BMI ≥ 30	8 (7%)	13 (12.1%)	17 (10.4%)
**Ethnicity (%)**			
White	45 (78.9%)	81 (75.7%)	126 (76.8%)
Asian	5 (8.8%)	24 (22.4%)	29 (17.7%)
Other	7 (12.3%)	2 (0.0%)	0 (0.0%)
**Level of Education**			
Year 12 or lower	22 (38.6%)	37 (34.6%)	59 (36%)
Trade or diploma	22 (38.6%)	22 (20.6%)	44 (26.8%)
Bachelor degree or higher	13 (22.8%)	48 (44.9%)	61 (37.2%)
**Food group serves median (IQR)**			
Fruit serves (150g)	0.6 (0.2–1.5)	0.8 (0.3–1.4)	0.8 (0.3–1.4)
Vegetable serves (75g)	1.6 (1.0–2.4)	1.9 (1.2–2.5)	1.8 (1.2–2.4)
EDNP food serves	3.2 (2.1–4.6)	2.9 (2.0–4.1)	3.0 (2.0–4.2)
SSB	0.4 (0.0–0.9)	0.3 (0.0–0.6)	0.4 (0.0–0.7)
Alcohol serves	0.0 (0.0–1.0)	0.0 (0.0–0.8)	0.0 (0.0–0.8)
Total EDNP food & beverages ^1^	4.4 (2.8–6.6)	3.9 (2.5–5.1)	4.1 (2.5–5.7)

^1^ Total energy-dense nutrient poor (EDNP) food group serves includes EDNP foods, sugar-sweetened beverages (SSB) and alcohol.

**Table 2 nutrients-11-00435-t002:** Comparison of perceptions for intervention group participants (n = 143) regarding the text message dietary feedback.

Statements Regarding the Dietary Feedback Text Messages	Responses, n (%)		
	Strongly Agree or Agree	Neither Agree or Disagree	Disagree or Strongly Disagree
**The text messages on my diet:**			
Told me things I did not know about my diet and what I eat	57 (39.9%)	39 (27.3%)	47 (32.9%)
Told me things about my diet I already knew	18 (12.6%)	32 (22.4%)	93 (65.0%)
Were useful in helping me to understand my diet ^1^	88 (61.5%)	35 (24.5%)	20 (14.0%)
Helped to motivate me to change my diet	74 (51.7%)	36 (25.2%)	33 (23.1%)
Made no difference to my motivation to change my diet ^1^	66 (46.2%)	34 (23.8%)	43 (30.1%)
Made me feel better about my diet	22 (15.4%)	61 (42.7%)	60 (42.0%)
Made me feel worse about my diet	43 (30.3%)	51 (35.9%)	48 (33.8%)
**Made me think:**			
About the foods I eat but only for a short while	87 (60.8%)	19 (13.3%)	37 (25.9%)
About how much fruit I eat	96 (67.1%)	19 (13.3%)	28 (19.6%)
About how much vegetables I eat	102 (71.3%)	18 (12.6%)	23 (16.1%)
About how much junk food I eat ^2^	93 (65.0%)	23 (16.1%)	27 (18.9%)
About how much alcohol I drink	22 (20.0%)	38 (34.5%)	50 (45.5%)
About how much soft drink and sugary drinks I have ^3^	46 (38.3%)	30 (25.0%)	44 (36.7%)

^1^ Statistically significant (*p* < 0.05) difference between men and women. ^2^ Junk food = EDNP foods. ^3^ soft drink and sugary drinks = SSB.

**Table 3 nutrients-11-00435-t003:** Logistic regression analyses adjusted for age and sex relating participants’ positive perception on text message dietary feedback to the actual change in food group serves (by 0.5 serves). Effects are represented as odds-ratio and associated 95% confidence intervals. Odds-ratio represents the increase in the likelihood of participants who agreed compared to those who disagreed, that the text messages made them think about how much they ate and their actual intake.

	Actual Change in Food Group Serves (by 0.5 Serve)					
Perception Questions ^1^	Increased Vegetables	Decreased EDNP Foods	Increased Fruit	Decreased SSB	Decreased Alcohol	Decreased Total EDNP Foods and Beverages
Vegetables	4.28 (1.76–10.39) *p* = 0.001	2.78 (1.28–6.04) *p* = 0.010	2.41 (1.10–5.27) *p* = 0.027	-	-	2.39 (1.1–5.10) *p* = 0.024
Fruit	-	1.94 (0.93–4.08) *p* = 0.079	-	2.34 (0.85–6.28) *p* = 0.097	-	2.66 (1.27–5.60) *p* = 0.010
EDNP food	-	2.47 (1.12–5.260) *p* = 0.025	-	-	-	1.93 (0.92–4.06) *p* = 0.083
SSB	-	-		-	-	2.05 (0.01–4.63) *p* = 0.084
Alcohol	-	-	-	-	4.59 (1.53–43.7) *p* = 0.006	-

^1^ Perception questions were undertaken at the completion of the intervention where those who agreed compared with those who disagreed (Reference group): Vegetables: Made me think about how much vegetables I eat. Fruit: Made me think about how much fruit I eat. EDNP foods: Made me think about how much junk food I eat. SSB: Made me think about how much soft drink and sugary drinks I have. Alcohol: Made me think about how much alcohol I drink.

**Table 4 nutrients-11-00435-t004:** Open-ended responses of young adults regarding dietary feedback.

Themes	Examples of Comments	
**What participants liked most about the dietary feedback text messages**		
Made me think	“Just a reminder and made me think about eating fruit for a snack rather than something else” (female).	“interesting comments … made me think momentarily about my diet but I continued old habits almost straight away” (female).
Constructive, helpful and useful	“I appreciate having a greater depth of consciousness as to what healthy food I can eat & found your directions helpful” (male).	“It was constructive. Helped to change my eating ways” (female).
Encouragement or motivation	“It was a wakeup call as to the horrible truth which is my poor diet choices! It motivated me to think more about changing my diet however time has certainly been a restriction” (female).	“Wasn’t all criticism, there was encouragement also” (male).
Personal, specific to me	“I loved the data given about my personal diet habits. They made me realize how much fruit and veg I SHOULD be eating” (male).	“specific to me not just a guideline in a magazine”(female).
Shocked and surprised	“I liked knowing that I ate a minimal amount of fruit and veg as it shocked me into making dietary changes. I’m not sure how long lasting these changes were though” (female)	“I was surprised that my fruit + veg consumption was lower than 2 fruit + 5 veg. I have tried to increase this since” (female).
**What participants liked least about the dietary feedback text messages**		
More detail	“It was very general feedback. It would have been good to have feedback more specific to the individual (e.g., Daily energy expenditure etc.)” (female).	“It wasn’t very comprehensive, compared to the data collected! I expected a much more detailed analysis of what I should/had eaten for my age, weight, sex etc. Not just fruit veg and junk” (female).
Confusing or vague	“Junk food recommendations a bit vague ’try only eat these foods sometimes’ something like ’try not to have more than 4 serves a week’ (eg) would have been more helpful” (female).	“The description of junk food was confusing. I did not understand what it meant” (female).
**What else participants would have liked feedback on**		
Portion size or quantity	“Overall quantity of food eaten - whether I should be eating more or less” (female).	“portion sizes, additional critiques about small changes that could be made” (male).
More about me	“more about MY diet” (male)	“Potentially specific things I need like iron and calcium. Important for my health condition” (female).
Enough protein	“Carb, protein, GI, energy levels for my own body, or e.g., Meal 32 was great! Because..” (female)	“protein (enough? Too much?), variety of my diet, GI or sustained energy tips” (female).
**If the text messages were sufficient**		
More feedback	“Text messages were good, however, an email with more personal findings would have been beneficial” (male).	“Maybe a bit more detailed feedback via email would be good to help ensure the things that I was doing well and continue to provide more feedback on areas I could improve ie healthier options” (female).
Short and to the point	“I liked that it was short and to the point and gave great handy tips” (female).	“It was to the point and focused on the important aspects of my diet that needed improvement. Any longer would have been a hassle to read.” (female).

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
