# Peer review of "Image-Based Dietary Assessment and Tailored Feedback Using Mobile Technology: Mediating Behavior Change in Young Adults"

_nutrients, 2019, doi:10.3390/nu11020435_

Round 1

Reviewer 1 Report

The research has promising value, and could be implemented in many health-related areas requiring behaviour change. 

Many factors could constitute a poor diet, the authors should add a definition of what a poor diet is. Perhaps some statistics around F and V intake would illustrate the argument. 

The authors cite research on lines 54-56 that adults aged 18-24 have the least F and V intake etc. The sample in the study however includes adults 18-30, it is unclear why the study has used this age range and not justified at present.  

Section 2.1 requires much more detail. 

Section 2.2 requires supplementary information, e.g., examples of the standard message template, examples of the personalised feedback. 

Section 2.6 requires more detail, specifically on the conduct of qualitative analysis. At present, this describes both a mix of thematic and content analysis approaches. The authors should explain the analysis in more detail and justify the qualitative approach used. 

Table 1 - data is split by gender, however subsequent analysis is not. The authors should justify why this is done for the decriptive statistics only and not the remainder of the analysis or group the data and provide the number of females and males respectively.

Further to point 6, data could be analysed by BMI, ethnicity, and education level. There may be some insightful findings, particulary as research tells us that F and V intake is less in lower socioeconomic groups and among different ethnicities. Results could provide direction for future research. 

The authors need to highlight where statistical significance is in the tables. At the moment, the tables are hard to understand at first glance. 

Table 4 is too lengthy, one quote per theme is sufficient to illustrate. 

The discussion is too general and requires more thought and detail. 

The discussion introduces psychological concepts (cognitive dissonance/ self-determination theory) not discussed in the introduction. These need to be discussed in the introduction if relevant to the research. At present, lines 244 - 272 present more of an introduction. This section requires restructuring. 

The importance of the research and application of the research findings are not discussed. 

Author Response

Response to Reviewer 1 Comments

We thank the reviewer for identifying that the research has promising value, and could be implemented in many health-related areas requiring behaviour change. Please note both line and reference numbers will refer to the revised manuscript.

Point 1: Many factors could constitute a poor diet, the authors should add a definition of what a poor diet is. Perhaps some statistics around F and V intake would illustrate the argument.

Response 1:

The reviewer has raised an important point. The evidence for poor diet has increased from single nutrients to the role of foods, such as fruit and vegetables and wholegrains.

We have provided an additional reference and modified the text to make this point more clearly (lines 49-53):

Poor diet is associated with increased risk of diseases such as cardiovascular disease; type 2 diabetes; and some cancers, with obesity a mediating factor [1,2]. Australian guidelines recommend regular consumption of diets high in fruit and vegetables, limiting consumption of alcohol and avoiding energy dense nutrient poor (EDNP) foods high in added sugars, saturated fat and sodium to reduce the risk of diet-related diseases [2].

Point 2: The authors cite research on lines 54-56 that adults aged 18-24 have the least F and V intake etc. The sample in the study however includes adults 18-30, it is unclear why the study has used this age range and not justified at present

Response 2:

Thank you for pointing this out as it was a typo. The Australian Bureau of Statistics report the data for young adults as 19-30 year olds. This age group has the lowest consumption of fruit and vegetables and highest intake of energy-dense nutrient-poor foods compared with all other adults. We have removed the age range to improve the readability and added text to justify the age range.

We have modified the text in paragraph 1 (lines 57-65):

Compared with other age groups, Australian young adults consume the least amount of fruits and vegetables and the most alcohol and EDNP foods and beverages with more than one-third (35%) of total energy coming from EDNP food and beverages [4,5]. Consequently, young adults are gaining weight more rapidly than any other age group, leading to an earlier onset of overweight and obesity than in previous generations putting them at a greater risk of preventable diet-related disease [6]. Young adults, a group in transition from adolescence to adulthood, represent an important target group for improving dietary habits and preventing weight gain.

Point 3: Section 2.1 requires much more detail.

Response 3:

We have provided more detail on the participant recruitment, which we believe is what the reviewer is asking. As the protocol ([24] Kerr et al. 2012) and main study outcomes ([9] Kerr et al. 2016) (have been previously published we did not want to replicate these details in this paper and have therefore provided a brief overview of the study with specific citation of the other papers to allow readers to locate large amounts of detail there (Lines 112-117):

The study protocol and trial outcomes have been published previously [9,24]. Participants were selected from 57 suburbs within the Perth metropolitan area in Western Australia to provide representation across socio-economic status through the Commonwealth electoral roll, a compulsory enrolment system for Australian adults. After receiving a letter of invitation, those who wished to take part underwent eligibility screening either by telephone or the study website.

Point 4: Section 2.2 requires supplementary information, e.g., examples of the standard message template, examples of the personalised feedback.

Response 4:

A detailed description of the construction and content of the text messages have been reported in [8] Kerr et al. 2016 so we were careful not to duplicate this information. However we can see that more detail is needed to help readers understand the findings of the study. 

The following text has been added (lines 144 – 156) :

Briefly, for the fruit and vegetable messages, participants received a message based on three levels of intake. Low intake was considered 0 to < 3.5 servings of fruits and vegetables; medium was 3.5 to < 7 servings of fruits and vegetables; and at least 2 servings of fruits and 5 servings of vegetables per day met the recommendations. For example, for a low intake, “Hi Jane, it’s Kate from CHAT with your feedback. So how did you score? Ave fruit serves = 0. 5, ave veg serves = 0.5. Your fruit serves varied from 0-1, veg from 0-1 over 4 days. What’s the goal again? 2 fruit and 5 veg a day. You can only go up from here!”. For EDNP serves, participants received a message based on three levels of intake and personalized to the participant’s dietary intake. For EDNP serves of 3 or more per day, the message was personalized with key sources of EDNP serves identified from the mFRTM. For example, “Hi Pete, It’s Kate from CHAT with your junk food score. Ave serves = 5, varying from 4-8 over 4 days. Junk foods are fatty or sugary foods that are high in calories. So try to only eat these foods sometimes and in small amounts. Could you try eating less fatty foods e.g. pies and sweet biscuits?”. A low intake of EDNP serves (0 – 3 serves daily serves) included the text “looks like you are on the right track”.

Point 5: Section 2.6 requires more detail, specifically on the conduct of qualitative analysis. At present, this describes both a mix of thematic and content analysis approaches. The authors should explain the analysis in more detail and justify the qualitative approach used.

Response 5:

We have revised this section and have moved content under 2.4 (lines 167-186). Thematic evaluation of open-ended responses is appropriate as the aim of the current study was to report on young adults’ experiences of receiving the dietary feedback and to determine whether those experiences were associated with positive improvements in dietary intake.

All free text responses were read several times by DK, CS and CP to become familiar with the data. The questions of the survey directed the reading of the responses so a deductive approach was used. Descriptive labels were used to organise information into themes. As all the available data was written in response to specific questions about the study, there was no opportunity for inductive analysis or discovering subthemes. Coding was conducted by three researchers (DK, CS, CP) independently. There was high agreement between coders but any disagreements were highlighted for discussion and committed to a theme. Differences were mainly about the name used to assign a theme e.g. misunderstanding or confusion.

We have modified the text as follows (lines 180-186):

These open-ended comments and free text responses were imported verbatim into NVivo 12. Qualitative data were coded and patterns identified using thematic analyses by three researchers (DK, CS, CP) independently. Descriptive labels were then applied to categorize information into themes. The researchers met and reviewed together the findings to confirm key themes. Discussion and revision of themes were made where required.

Point 6: Table 1 - data is split by gender, however subsequent analysis is not. The authors should justify why this is done for the descriptive statistics only and not the remainder of the analysis or group the data and provide the number of females and males respectively.

Response 6: Table 1 is a descriptive table to provide details of participants enrolled in the study. The breakdown by gender was purely for further describing the sample.

Gender was considered as a covariate in the subsequent logistic regression analyses, as stated in the Methods section. The title for Table 3 has been edited to indicate the adjustment.

Point 7: Further to point 6, data could be analysed by BMI, ethnicity, and education level. There may be some insightful findings, particularly as research tells us that F and V intake is less in lower socioeconomic groups and among different ethnicities

Response 7:

BMI, ethnicity, education level, and socioeconomic groups were also considered as covariates within the analyses, but these were found to be non-significant (p>0.1). In the interest of parsimony, the simplest multivariable models, which explain most of the variance, have been presented in the paper.

We have added more detail in the analyses section (lines 194-196):

Preliminary analysis revealed that BMI, ethnicity, education level, and socioeconomic status were not associated (p>.1) with change in food group serve, hence were not included in the multivariable model.

Point 8: Results could provide direction for future research.

Response 8:

We agree this important. We have addressed future research in the discussion (line 351-354):

To further understand motivations towards changing dietary behaviors, future studies should study these motivational processes when planning dietary interventions. This should include examining autonomy for nonadherence in young adults whilst being respectful of their dietary choices [34].

Point 9: The authors need to highlight where statistical significance is in the tables. At the moment, the tables are hard to understand at first glance.

Response 9: We have modified the Tables to improve the clarity.

Point 10: Table 4 is too lengthy, one quote per theme is sufficient to illustrate.

Response 10: We have revised Table 4 as suggested.

Point 11: The discussion is too general and requires more thought and detail. The discussion introduces psychological concepts (cognitive dissonance/ self-determination theory) not discussed in the introduction. These need to be discussed in the introduction if relevant to the research. At present, lines 244 - 272 present more of an introduction. This section requires restructuring.

Response 11: We have made changes to the introduction and discussion and believe we have improved the readability and flow. In the introduction we have introduced behavior change theory, including tailoring (lines 66 – 85) and attempted to make the discussion tighter. Further the changes to the introduction and methods we believe the discussion is framed differently and as such is tied differently to concepts and therefore less general.

Point 12: The importance of the research and application of the research findings are not discussed.

Response 12: Thank you for pointing this out. We have also addressed this in the conclusion. We have modified the text to illustrate what we believe are the strengths of our study (Lines 342-346).

Most studies to date, have based participant dietary feedback on short questions rather than more detailed dietary records [41]. A major strength of this study is the collection of dietary intake using a 4-day mFRTM which provided a more detailed and personalized measure of dietary intake. This enabled an evaluation of whether young adults’ experiences of receiving the dietary feedback were associated with positive improvements in dietary intake.

Reviewer 2 Report

The presented work raises an interesting topic that can be of great importance in the dietary education of fashionable adults. Text message dietary feedback is a simple but extremely understandable educational tool for adults.

1.      Participant recruitment requires supplementation, how were the participants of the study resolved, whether they expressed their written consent to participate in the study. There is definitely a lack of division of study participants due to their place of residence: village, city.2.      The obtained data presented in Table 1 should be supplemented with statistical significance for the examined variables characterizing the study participants.3.      Tables 2 and 3 are very illegible. in table 2 it is necessary to introduce statistical significance, which will make it easier for the reader to interpret the obtained results. Table 3, however, requires a total reconstruction. Statistical significance should be marked with * for different p-value values.4.      The authors analyzed BMI, ethnicity and level of education of participants but these data were not later used in statistical analysis. It is necessary to look for dependencies if these variables have an impact on the positive reception of the text message dietary feedback. Did people with higher education in the same way reduce the share of SSB or EDNP food in the daily food ration as people with primary education. It is necessary to supplement the research results and present how the variables examined influenced the choices of participants in the study.5.      Table 4 can be reduced.6.      The discussion should refer to the studied variables characterizing the population. It is necessary to refer to the difficulties of changes in the way of diet after dietary intervention in people with high BMI.7.      The discussion (verses 244-272) is too lengthy and does not bring any relevant information to the presented results.8.      The authors have not shown the strengths of their work.9.      Applications are written too generally, they do not show the results achieved.10.   In the list of literature, the shoulder of DOI numbers

Author Response

Response to Reviewer 2 Comments

We thank the reviewer for their positive comments regarding that the topic of our manuscript can be of great importance in dietary education. We have outlined our detailed response as below. Please note both line and reference numbers will refer to the revised manuscript.

Point 1: Participant recruitment requires supplementation, how were the participants of the study resolved, whether they expressed their written consent to participate in the study. There is definitely a lack of division of study participants due to their place of residence: village, city

 Response 1:

All participants signed an informed consent. This detail has been added to the manuscript (line 123).

We have also added more details on the study recruitment as follows (lines 113-117):

The study protocol and trial outcomes have been published previously [9,24]. Participants were selected from 57 suburbs within the Perth metropolitan area in Western Australia to provide representation across socio-economic status through the Commonwealth electoral roll, a compulsory enrolment system for Australian adults. After receiving a letter of invitation, those who wished to take part underwent screening either by telephone or the study website.

Point 2: The obtained data presented in Table 1 should be supplemented with statistical significance for the examined variables characterizing the study participants.

 Response 2:

Table 1 is a descriptive table to provide details of participants enrolled in the study. We don’t believe testing significance by gender is necessary or appropriate. 

Point 3: Tables 2 and 3 are very illegible. in table 2 it is necessary to introduce statistical significance, which will make it easier for the reader to interpret the obtained results. Table 3, however, requires a total reconstruction. Statistical significance should be marked with * for different p-value values

 Response 3:

The annotations for Table 2 and 3 have been edited to improve interpretation. Statistical significance is in the notation for Table 2. We have edited Table 3 to improve the clarity. We have included the specific p values along with the odds-ratio and confidence intervals in Table 3 as this is the appropriate way to display the data. We have added the following text to Table 3:

Odds-ratio represent the increase in likelihood of participants who agreed compared to those who disagreed, that the text messages made them think about how much they ate and their actual intake.

Point 4: The authors analyzed BMI, ethnicity and level of education of participants but these data were not later used in statistical analysis. It is necessary to look for dependencies if these variables have an impact on the positive reception of the text message dietary feedback. Did people with higher education in the same way reduce the share of SSB or EDNP food in the daily food ration as people with primary education. It is necessary to supplement the research results and present how the variables examined influenced the choices of participants in the study

 Response 4:

The results of the logistic regression were adjusted for age and sex and used to analyze the change in daily food group serves (by 0.5 serves) from baseline to follow-up with participants’ perceptions on whether the dietary feedback text messages made them think about consumption of vegetables, fruit, EDNP, SSB or alcohol. BMI, ethnicity, education level, and socioeconomic groups were also considered as covariates within the analyses, but these were found to be non-significant (p>0.1). In the interest of parsimony, the simplest multivariable models which explains most of the variance have been presented in the paper.

Point 5: Table 4 can be reduced

 Response 5: As suggested we have reduced Table 4.

Point 6: The discussion should refer to the studied variables characterizing the population. It is necessary to refer to the difficulties of changes in the way of diet after dietary intervention in people with high BMI

 Response 6:

We are aware of the difficulties in changing dietary behaviour. However our findings are unable to provide any evidence on whether there was any association with a higher BMI. As outlined above in point 4, BMI, ethnicity, education level, and socioeconomic groups were also considered as covariates within the analyses, but these were found to be non-significant (p>0.1). Therefore we have not reported these.

Point 7: The discussion (verses 244-272) is too lengthy and does not bring any relevant information to the presented results.

 Response 7:

We have revised the discussion. We have not reduced the length of the discussion, rather sort to make succinct points, as we believe the discussion is relevant and allows a full exploration of the findings.

Point 8: The authors have not shown the strengths of their work.

 Response 8:

Thank you for pointing this out. We have modified the text to illustrate what we believe are the strengths of our study (Lines 342-346).

 Most studies to date, have based participant dietary feedback on short questions rather than more detailed dietary records [41]. A major strength of this study is the collection of dietary intake using a 4-day mFRTM which provided a more detailed and personalized measure of dietary intake. This enabled an evaluation of whether young adults’ experiences of receiving the dietary feedback were associated with positive improvements in dietary intake.

Point 9: Applications are written too generally, they do not show the results achieved.

 Response 9:

We have revised the discussion to address this point. We have made changes to the introduction and discussion and believe we have improved the readability and flow. In the introduction we have introduced behavior change theory, including tailoring (lines 66 – 85) and attempted to make the discussion tighter. Further the changes to the introduction and methods we believe the discussion is framed differently and as such is tied differently to concepts and therefore less general.

Point 10: In the list of literature, the shoulder of DOI numbers

 Response 10:

Thank you for pointing this out. The author guidelines are unclear. We will add the DOI should the editor require them.

Round 2

Reviewer 2 Report

Changes made after the review allow for a better understanding of the topic being discussed. The modification of table 3 is very good and allows better interpretation of the results obtained.